# In situ stable crack growth at the micron scale

Giorgio Sernicola [1], Tommaso Giovannini[2], Punit Patel[3], James R. Kermode[3], Daniel S. Balint[2], T. Ben Britton[1] & Finn Giuliani[1,2]

Grain boundaries typically dominate fracture toughness, strength and slow crack growth in ceramics. To improve these properties through mechanistically informed grain boundary engineering, precise measurement of the mechanical properties of individual boundaries is essential, although it is rarely achieved due to the complexity of the task. Here we present an approach to characterize fracture energy at the lengthscale of individual grain boundaries and demonstrate this capability with measurement of the surface energy of silicon carbide single crystals. We perform experiments using an in situ scanning electron microscopy-based double cantilever beam test, thus enabling viewing and measurement of stable crack growth directly. These experiments correlate well with our density functional theory calculations of the surface energy of the same silicon carbide plane. Subsequently, we measure the fracture energy for a bi-crystal of silicon carbide, diffusion bonded with a thin glassy layer.

[1] Department of Materials, Imperial College London, Kensington, London SW7 2AZ, UK. [2] Department of Mechanical Engineering, Imperial College London, Kensington, London SW7 2AZ, UK. [3] Warwick Centre for Predictive Modelling, School of Engineering, University of Warwick, Coventry CV4 7AL, UK. Correspondence and requests for materials should be addressed to G.S. (email: g.sernicola13@imperial.ac.uk)

Building the next generation of advanced engineering technologies is often limited by the quality and performance of available materials. For applications such as space telescopes, atmospheric re-entry vehicles, and medical implants, materials with high hardness, high thermal stability, high strength, and low density are required. Ceramics have these attributes; yet their widespread use is restricted due to their poor resistance to fracture, often leading to unpredictable and catastrophic failures. If toughness could be improved, through mechanistically informed grain boundary engineering and precise process control, then fabricated components could be used in ever more demanding environments. Successful manufacture of these improved ceramics relies on understanding which grain boundaries are preferred, i.e., sampling- preferred boundary misorientations and interface chemistries, in order to enable the creation of tailored microstructures that optimize fracture performance. This approach requires targeted experimental tests of individual microstructural features to provide mechanistic insight of which grain boundaries are preferred and why. Furthermore, these tests may assist rationalization of existing experiments, for instance in silicon carbide (SiC) it has been shown that chemistry of individual interfaces can have a strong influence on the fracture toughness[1, 2].

At present, there are many standard tests that enable precise measurement of the fracture properties of brittle materials at a macroscopic lengthscale[3, 4], yet few available tests that enable direct measurement of individual grain boundaries. Macroscopic tests are very successfully used to understand bounds on component performance and to compare and contrast ceramic-processing strategies, but as they test polycrystalline aggregates it is very difficult to understand the role of specific microstructural features. Achieving a step change in the performance and utilization of advanced ceramics requires new insight at the microstructural lengthscale, as failure of ceramics is controlled often by the weakest microstructural link.

Development of site-specific micromechanical testing of ceramics has advanced significantly in the last 5 years, using either photolithography or focused ion beam (FIB) machining, enabling preparation of microscopic test specimens in precisely specified locations and opening the door to high spatial resolution mechanical tests.

Typically, these test specimens are loaded using a nanoindentation platform with different tip geometries according to the test design. These geometries include: single cantilever bending[5–11], double cantilever beam (DCB) compression[12], clamped beam bending[13, 14] (see Fig. 1a–c), and pillar splitting[15]. The philosophy of performing high spatial resolution testing, as employed with these geometries, enables assessment either of the local fracture properties of single grains or grain boundaries, or of samples that are inherently small in volume, e.g., coatings. The key limitation of existing approaches is that they typically make use of a load-controlled ramp to actuate the indenter and follow crack opening indirectly, using changes in load-displacement data both to identify the crack nucleation and fracture load to be used in fracture toughness measurements. This is problematic if the loading system is compliant or stores significant energy, and load control exhibits an increasing energy release rate with crack growth therefore inherently does not lend itself to stable fracture.

Unfortunately, an unstable loading geometry limits evaluation of fracture energy to extraction of a single value. This issue is compounded by warranted concerns about the effects of notch manufacture at this small lengthscale, such as the introduction of FIB-induced damage in the region surrounding the notch[16].

In light of these previous geometries[8], it would be ideal to have test samples with geometrical features enabling stable crack growth beyond any damaged region, in order to measure fracture toughness as the crack evolves and to overcome limitations imposed by FIB-induced damage. It would be also useful to have freedom in the positioning of the notch combined with a relatively simple sample geometry, thus facilitating sample fabrication and fracture or surface energy analysis. Furthermore, a minimization of the effect of frame compliance and friction between the indenter and the sample would make evaluation of the measured energy easier.

In this manuscript, we choose to test a DCB geometry, fabricated using FIB, and load this with a wedge in direct displacement control in situ within a scanning electron microscope (SEM). This geometry is similar to the classic design described by Lawn[17], and utilizes the displacement of each cantilever beam and a simple calculation of the stored elastic energy within each beam that balances with the energy required to create new crack surface. This enables a direct measurement of the fracture or surface energy as the crack grows. As the experiment is performed in displacement control, the driving force decreases with crack length and therefore cracking is inherently stable[17–19]. Furthermore, in situ testing allows the atmosphere to be controlled, reducing concerns of environmentally induced effects such as stress corrosion cracking[20]. In our test, stable crack growth is demonstrated through a prolonged displacement hold (between 150 and 300 s) with no observable increase in crack length. Surface energy measurements are demonstrated on single crystal SiC samples to validate the technique and these are compared with density functional theory (DFT) calculations. Experimentally, the approach is subsequently used to explore the fracture of SiC bi-crystals adhered together with silica (manufactured using diffusion bonding). In summary, this work presents a method to measure the fracture energy evolution with crack length at the microscale in purely brittle materials, with the benefits of a high sampling rate, stable crack growth, and a controlled environment.

## Results

**Fracture energy measurement.** Experiments were performed on single crystal 6H-SiC and bi-crystal SiC, which was diffusion-bonded with a silica glass interface layer. The DCBs were fabricated from the ceramic by FIB machining to create the shape shown in Fig. 1d. Mechanical testing was performed in situ in a SEM, which provided high spatial and temporal resolution imaging of the loading and fracture processes. This proved beneficial for the alignment of the sample and loading geometry, as well as providing direct observations of the fracture under load.

A wedge sliding through the central trough causes beam bending of both cantilevers, and the elastic energy stored within the beams is available to drive crack advance. By applying simple beam theory, it is possible to calculate the energy spent in beam bending, in terms of the beam displacement, beam geometry, and the elastic stiffness of each cantilever. The elastic strain energy, $U_M$, stored in each beam per unit depth is given by Euler–Bernoulli beam theory as:

$$U_M = \frac{Ed^3\delta^2}{8a^3};$$ (1)

where $E$ is the elastic modulus, $d$ the beam width, $\delta$ the maximum displacement, and $a$ the crack length (Fig. 1d).

The strain energy release rate (energy per unit area), $G$, is given by the Griffith criterion as:

$$G = -\frac{dU_M}{da} = \frac{3Ed^3\delta^2}{8a^4}$$ (2)

The DCB system can be approximated as two individual clamped end-loaded cantilevers (see Supplementary Fig. 1), where

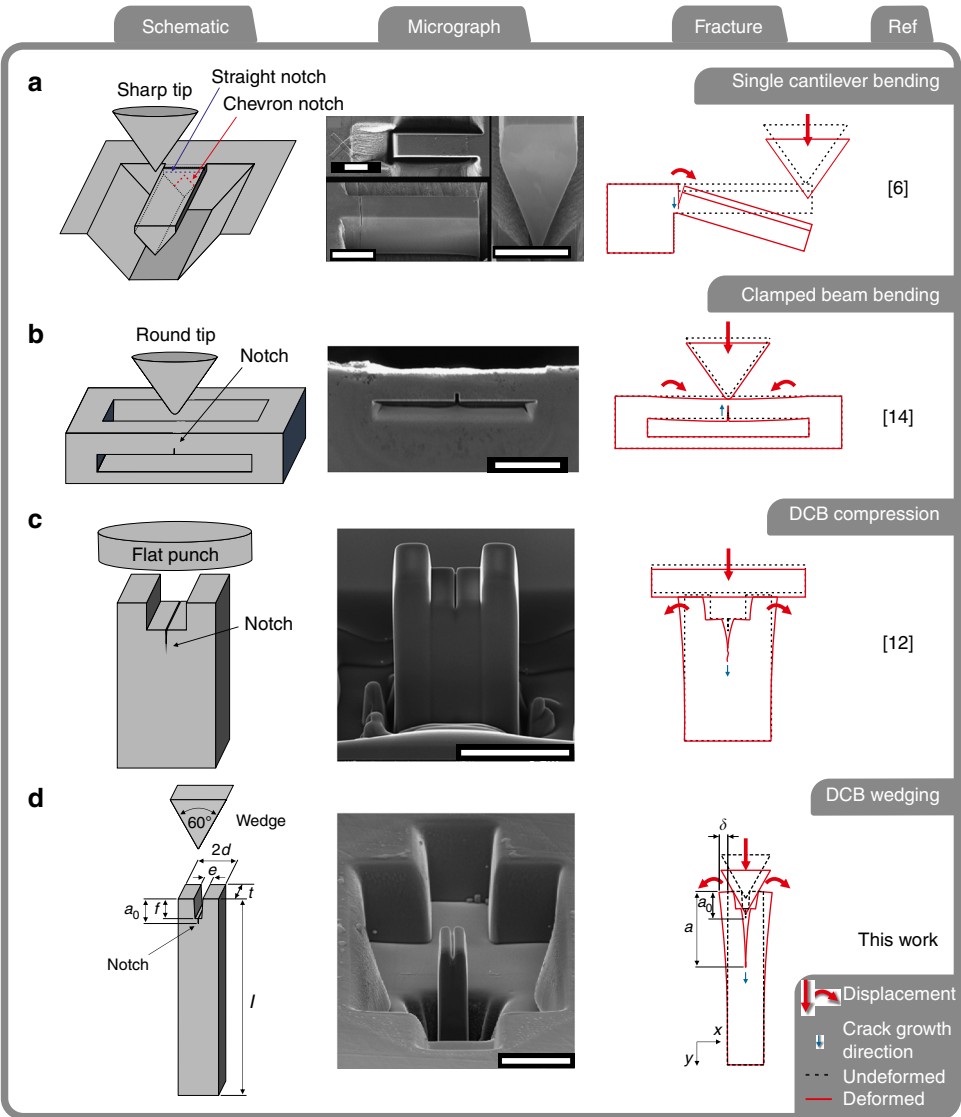

**Fig. 1** Loading geometries employed for micromechanical fracture testing of interfaces. **a** Single cantilever bending geometry with pentagonal cross-section and loaded with a sharp indenter. *Rectangular cross-section* is only possible on a 90° edge[7, 8] and a pentagonal cross-section is toward the sample interior. Straight and chevron notch geometry sections are represented with *blue* and *red dashed lines*, respectively. Straight notches provide inherently unstable fracture, whereas the chevron notches enable a stable crack growth for short cracks[8, 11]. Dimension measurements of notch and beam geometry are performed after the test[5–7]. *Scale bar*: 10 μm. **b** Schematic of a double clamped beam bending geometry loaded with a blunt tip. The beam is FIB milled on a 90° sharp edge and a notch at the *bottom edge centre* of it[14]. Stable crack growth is supported but freedom of positioning of the notch is limited[8]. *Scale bar*: 40 μm. **c** DCB geometry loaded with a flat punch. Stable crack growth is achieved. Friction between flat punch and sample, and compressive stress on struts need to be taken into account[12]. *Scale bar*: 3 μm. **d** The wedge loaded DCB geometry used in this work. Dimensions are: 10 < *l* < 15 μm, *2d* ~ 2 μm, *t* ~ 5 μm, *e* ~ 400 nm, 1 < *f* < 2 μm, and 1.5 < $a_0$ < 2.5 μm. The beam displacement δ at the contact point between the wedge and the DCB, and the crack length *a*, as shown, allow to measure the energy stored in the beam once beam width and stiffness are known. *Scale bar*: 5 μm

the clamp position is at the crack tip and the loading point is at the contact point between the wedge and the beam.

For the initial crack growth, the beams are shorter than the approximation required for simple beam theory to hold (i.e., $d \gg a$ when the crack is short); therefore, the model is extended using the linear elasticity solution for the short crack configuration, which naturally includes the shear contribution. This results in an energy release rate, $G$, for each beam (see Supplementary Note 1 for the derivation):

$$G = \frac{3Ed^3\delta^2}{8a^4} + \frac{3E(1+\nu)d^5\delta^2}{8a^6} = \frac{3Ed^3\delta^2}{8a^4}\left[1 + (1+\nu)(d/a)^2\right];$$

(3)

where $\nu$ is Poisson's ratio; note that Eq. (3) simplifies to Eq. (2) when $a \gg d$. The total energy release rate can be calculated using Eq. (3), summing the $G$ values calculated for each beam independently (which are not necessarily displaced by the same amount δ).

Measurement of beam width, beam displacement, and crack length was performed using custom MATLAB scripts from SEM image frames captured within the in situ experiment.

Initial registration of all frames was performed using image cross-correlation, keeping a point away from the bending DCB fixed within the frame. This reduces the effects of image drift caused by sample or nanoindenter compliance and sample charging under the electron beam.

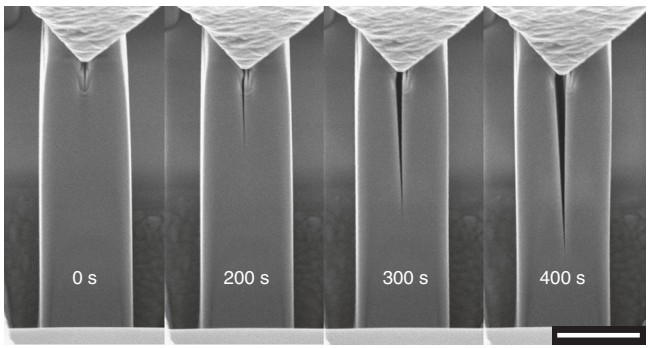

**Fig. 2** Slow stable crack growth. A sequence of frames extracted from the *video* recorded during one of the in situ tests. The *wedge tip* at the *top* can be seen moving down as the test advances, causing the opening of the DCB; the crack slowly grows for a few microns over a total time of more than 350 s until the tip displacement is stopped and held in position. *Scale bar*: 2 μm

Next, measurements of maximum displacement $\delta$ were performed using a similar cross-correlation procedure to track the DCB edge position at the contact point with the wedge for each frame, with respect to the position within the first frame, i.e., when the beam was still unloaded.

The coordinates of the crack tip position were selected by hand, augmenting contrast within the script to make this easier. Each cantilever beam width was measured using frames toward the end of the test, when the crack was longer. As the nanoindenter sits on the SEM stage with its indentation axis tilted 30° with respect to the horizontal plane, all the measurements along the vertical direction of the image were corrected for foreshortening.

For the analysis, a Monte Carlo-based error propagation script was used. Variables for the analysis with known errors were adjusted independently to have a Gaussian distribution with a known standard deviation. The mean of each distribution was the measurement for each test and the standard deviations based upon experimental measurement error. The variables and distributions are listed in the Supplementary Table 1 and Supplementary Fig. 2, while the result of the error propagation analysis on one of the SiC DCBs is presented in Supplementary Fig. 3.

In the single crystal SiC DCB-labeled SC 4 the crack initiated at the pre-notch and grew without any bursts until the test was terminated and the tip retracted after an ~2 μm crack growth. In the other three single crystal SiC DCBs, the crack initiated ~100 nm on the left of the FIB-milled notch, indicative of a slight asymmetry in the loading geometry. Once the DCB was loaded, the crack propagated straight and its growth was stable. This is shown in the sequence of frames extracted from the video recorded during the test of SC 2 (Fig. 2) and in the video of the test on SC 3 (Supplementary Movie 1); the test was stopped when it reached a crack growth of ~4 μm.

Figure 3b shows the deflection of the two beams of the DCB SC 3 plotted against time from the start of the test measured in pixels, and then converted to microns, as obtained through image cross-correlation. After an initial period of energy storing with no lateral movement of the beams, the data show a linear displacement of both cantilevers at the contact point.

Although the frames were registered for overall movements independent from beam bending as explained in the experimental methods, an equal and opposite movement of the two beams is evident in the first portion of the plot in Fig. 3b, indicating a movement of the DCB under the initial load. The bending starts after ~ 200 s and continues to increase linearly until the tip displacement is stopped and held in position. The cross-correlation records negligible movement of the beams during this

stage; an increase in beam displacement was only recorded when the tip displacement was resumed.

For the purpose of fracture energy measurements, only the time interval during which the crack growth was observed was taken into account (see crack growth interval indicated in Fig. 3b). In order to reduce noise we performed a linear fit of the data for the energy measurement calculation.

The history of crack growth with time (one plot is shown in Fig. 3c), although presenting small jolts, is relatively steady and is highly controlled in all the DCBs tested. This was demonstrated in three of the tests (see Supplementary Movie 1 for the test on SC 3) in which, after an initial crack propagation, the wedge tip was held still for 5 min (Fig. 3a) with consequent crack arrest until the tip displacement was resumed. Similarly to the lateral displacement data, crack length against time data was fitted to a third degree polynomial to reduce noise prior to use in the fracture energy calculation. The crack speed measured as an average from start to end is in the range between 25 and 29 nm s$^{-1}$ for all the single crystal DCBs (this compares with a loading rate of between 1 and 2 nm s$^{-1}$).

Using the beam deflection, crack length, and beam thickness as measured from the test images, along with an elastic modulus of 480 GPa and Poisson's ratio of 0.18, the fracture energy data shown in Fig. 3d and Table 1a were obtained. Given the anisotropy of SiC, in our calculations the elastic modulus used is calculated using the values of elastic constants from Landolt-Börnstein[21] and is calculated for the orientation parallel to the (0001) plane. Similarly, the appropriate Poisson's ratio was extracted for the two mutually orthogonal directions to the (0001) plane.

This same method has been used to measure the fracture energy of the glassy interfaces in the three bi-crystal DCBs. For these experiments, the crack started tens of nanometers on a side of the milled notch (that acts as a stress concentrator) for all the tested specimen and proceeded constantly for ~ 2 μm until the tip displacement was stopped and retracted. Given the small volume of the interface compared to the SiC, the same Young's modulus and Poisson's ratio used for the single crystal SiC were used to measure the fracture energy of the interfaces (as the majority of the elastic energy is stored within the SiC beams).

Our results are summarized in Table 1, along with our DFT calculations (see Methods section) of the surface energy for 6H-SiC and previous literature results for the similar 3C-SiC{111} surface[22, 23]. For completeness, we calculated the respective fracture toughness values (see Table 1) through $K_{Ic} = \sqrt{E_\perp G_c}$, where $E_\perp$ is the elastic modulus perpendicular to the (0001) plane (taken as 554 GPa for the current experimental work, using the values of elastic constants from Landolt-Börnstein[21]). Given the small value of Poisson's ratio in this case (of order of 0.08 for $\nu_{12}$), the difference between plane stress and plane strain assumptions is negligible.

**Asymmetries and other geometrical complications**. Symmetry of the DCB geometry would allow the total strain energy stored, whose rate of dissipation with crack length is equal to the fracture energy during stable crack propagation (see Eq. (2)), to be obtained by measurement of the strain energy stored in one of its beams and simply doubling it.

However, in practice, perfectly symmetrical systems are unlikely in experimental sample preparation and testing conditions at this lengthscale. Therefore, in this work each beam was analyzed independently (Fig. 4b) and the total energy term obtained by summing the two values of the left and right beams.

Asymmetry is likely caused by three factors. Firstly, the high lateral stiffness of the nanoindenter tip housing system, which

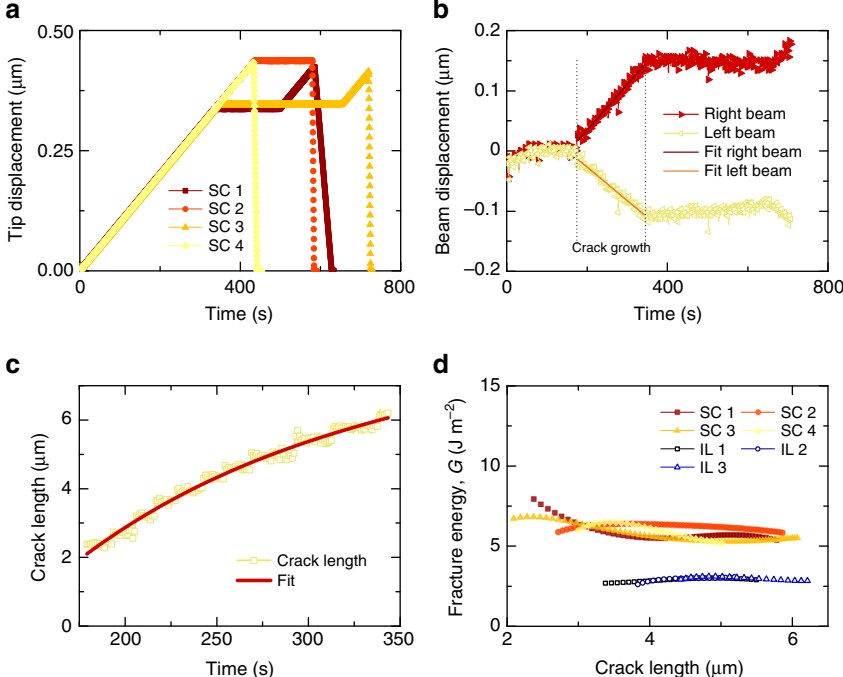

**Fig. 3** Data analysis of DCB cracking. **a** Tip displacements as set-up on the nanoindenter for four of the tests conducted on the single crystal (SC) SiC samples. **b** Displacement of *left* and *right* beams of the sample SC 3 measured by image cross-correlation of the edges during the total test length. *Dotted lines* are used to show the interval of time in which the crack growth was measured. The *curve* is fitted according to a third-order polynomial with $R^2 = 0.95$ for left beam and $R^2 = 0.96$ for right beam. **c** Crack length measured on sample SC 3 from the point of crack nucleation until the tip was stopped to hold position. Fitted data ($R^2 = 0.98$) were used for the fracture energy measurement to reduce the noise in the data. **d** Fracture energy value measured over crack growth for the four SC SiC samples and the three SiC bi-crystals with $SiO_2$ interlayers (IL), showing good reproducibility and a significant difference between the values of the two systems

makes any misalignment between the tip's central axis and the DCB's central axis hard to accommodate during the loading, is reflected directly in an asymmetric displacement of the beams. Secondly, a similar effect would be caused by a misalignment between the sample surface normal and the tip's displacement axis, as the two beams would be loaded at two different apparent angles by the wedge. Finally, a subtle difference was observed in beam thickness, due to difficulties in fabricating small samples (even with automated fabrication regimes as employed here).

For each of the three bi-crystal samples, cracks started tens of nanometers on the side of the milled notch. This is likely due to slight asymmetries in the overall loading geometry and the precise location of the interface.

These slight asymmetries require the energy stored in the cantilevers to be measured individually to avoid the fracture energy being significantly underestimated or overestimated. While this subtlety may not be obvious, in practice it is straightforward to implement asymmetric analysis with this in situ geometry. A comparison of the differences between crack growth measurements using asymmetric or symmetric analysis is presented in Fig. 4a.

**Short crack lengths**. In this work, the fracture energy results are measured, taking into account the non-negligible shear component that arises because the beam length is comparable with its thickness (Fig. 4b), i.e., when the crack is short. This low aspect ratio beam correction enables reliable measurement of the fracture energy value with variable crack length and demonstrates that the technique lends itself to the investigation of small regions of interfaces at short distances from the surface in real materials.

**Taper**. FIB milling normal to the sample surface is known to produce tapered final geometries; however, the analysis

presented in this work considers the cantilevers made of constant cross-section along its length. It is therefore important to design the milling steps to minimize the taper in the final geometry or alternatively the analysis must be performed using a more complicated elastic model, as afforded for instance with cohesive zone-based finite element models.

**Fracture energy or surface energy**. From the high-resolution SEM images the crack propagation in the single crystal SiC appears to create two new smooth edges with no evident deviation from a linear path. The crack tip is sharp and no plastic deformation is expected around it and likewise no toughening mechanisms are available in the material examined. Furthermore, it is worth noting that the energy values were measured far from the notch and so we do not expect to be affected by crack initiation processes or ion penetration damage, which is found to be <100 nm from the surface[24, 25].

These measured values of $5.95 \pm 1.79 \, J \, m^{-2}$ (measured as average of all the fracture energy measurement obtained with crack evolution and ± indicates the standard error as obtained by Monte Carlo-based error propagation, see Methods) thus are likely to be representative of twice the surface energy (i.e., $2\gamma$). Our DFT calculations estimate the surface energy to be $2\gamma = 8.15 \pm 0.44 \, J \, m^{-2}$, taking into account uncertainties arising from the existence of three inequivalent termination planes for 6H-SiC in the [0001] direction (Fig. 5), from the exchange-correlation functional used, and numerical approximations such as finite basis set (see Methods). The DFT estimate of the surface energy is slightly higher than the experiment.

Similarly, the values obtained by testing the bi-crystals potentially provide the surface energy of the SiC–glass–SiC interface. Wiederhorn[26] reported values of fracture surface energy

**Table 1 Comparison between experimental and simulated fracture/surface energy values of SiC and glass**

| Description | Lattice constant a/Å | Lattice constant c/Å | Surface energy* $2\gamma$ or Fracture energy** $G_c$/(J m$^{-2}$) | Fracture toughness $K_{Ic}$/(MPa m$^{0.5}$) | References |
|---|---|---|---|---|---|
| *(a)* | | | | | |
| 6H-SiC (micro-scale experiment) | – | – | $5.95 \pm 1.79$** | $1.80 \pm 0.26$ | Current work |
| 6H-SiC (macro-scale experiment) | – | – | $16-25$** | $3.3 \pm 0.2$ | (refs [37-39]) |
| 6H-SiC(LDA) | 3.05 | 15.02 | $8.58 \pm 0.04$* | $2.18 \pm 0.03$ | Current work |
| 6H-SiC(GGA-PBE) | 3.09 | 15.19 | $7.71 \pm 0.04$* | $2.01 \pm 0.03$ | Current work |
| 6H-SiC DFT combined | – | – | $8.15 \pm 0.44$ | $2.10 \pm 0.08$ | Current work |
| 3C-SiC(LDA) | 4.34 | – | $8.34$* | | (ref. [23]) |
| 3C-SiC(Tersoff screened) | 4.32 | – | $3.70$* | | (ref. [23]) |
| 3C-SiC(PBE) | – | – | $8.40$* | | (ref. [23]) |
| *(b)* | | | | | |
| Glassy interface of SiC bi-crystal | – | – | $3.35 \pm 1.16$** | | Current work |
| Glass of different compositions | – | – | $7.00-9.50$** | | (ref. [26]) |
| Theoretical estimate for silica glass | – | – | $1.00$* | | (ref. [26]) |

(a) Simulations and experimental results (see Methods for details) for the surface and fracture energies of 6H-SiC and 3C-SiC
(b) Theoretical estimate and experimental results for silica glass, and experimental results for the glassy interface of the SiC bi-crystal. The experimental values measured in the current work are the average of all the fracture energy measurement obtained with crack evolution. ± indicates the standard error as obtained by Monte Carlo-based error propagation. The respective fracture toughness values from the current work were calculated as detailed in Results and Methods sections

for glasses with different chemical composition in a range between 7.0 and 9.5 J m$^{-2}$ at 300 K, comparing them with a theoretical estimate of 1.0 J m$^{-2}$ for surface energy obtained by Charles[26]. The crack surfaces obtained by Wiederhorn showed edges with angles varying as much as 30° to straight path and these deviations will result in a larger work of fracture. Therefore, the lower value obtained in the present work falls reasonably between the theoretical and experimental values found in the literature.

## Discussion

In situ wedging of a DCB coupled with a displacement-controlled testing machine afforded a stable and relatively long crack growth at the micrometer lengthscale to be obtained.

The method was validated by fracturing single crystal 6H-SiC along the <a> direction, finding a value of the fracture energy of $5.95 \pm 1.79$ J m$^{-2}$. DFT calculations were performed on the same crystallographic plane as experimentally tested for the first time. While the calculated values were slightly higher than the experimental values, both measurements agree well when compared to previous measurements (Fig. 6). The consistency between experiment and DFT fracture energies for the single crystal case gives us confidence in the approach when moving to study more complex systems. By performing additional calculations (see Methods), we have been able to exclude significant effects of the surface termination layer, temperature, surface reconstruction, or hydrogen termination on the theoretical fracture energy.

Tests were then conducted on a SiO$_2$ interface of ~ 10 nm thickness used to bond two coupons of SiC. These results show a value of $3.35 \pm 1.16$ J m$^{-2}$, which compares with the range between theoretically estimated and experimentally measured fracture surface energy of glass in the literature.

The loading geometry employed proved that measurement of the evolution of fracture energy with crack length is possible at the microscale and without the use of load data; thus, reducing the effect of notch radius, ion damage, and frame compliance on the value found, which are often a concern in fracture testing at this lengthscale.

The method proposed introduces an alternative opportunity to study the fracture properties at the lengthscale of individual interfaces and microstructural units, thus opening up to the possibility of optimization of fracture properties of specific boundaries in ceramics with more complex microstructures.

## Methods

**Materials**. The single crystals were supplied by MTI Corporation. For the bi-crystal samples, two 6H-SiC single crystals were diffusion-bonded. The original SiC samples were coupons of $5 \times 5 \times 0.5$ mm. Prior to diffusion bonding a layer of tetraethyl orthosilicate was spin-coated to each of the component's surfaces. This was used as a precursor[27] for the silica glass allowing a SiC bi-crystal with a 10 nm grain boundary.

For diffusion bonding, the coupons were placed inside a vacuum furnace and kept at a pressure of $5 \times 10^{-5}$ Pa for the entire duration of the process. The chamber was heated to 1900 °C using a heating rate of 30 °C min$^{-1}$. An initial load of 6N (24 kPa) was applied normal to the area of contact between the two components. When the maximum temperature was reached, the load was increased to 2000 N (80 MPa) and held for 30 min. The sample was then cooled at a rate of 20 °C min$^{-1}$ to room temperature. The sample was removed from the furnace and mounted to present the interface vertically for subsequent mechanical testing.

**DCBs fabrication**. FIB milling was operated in a FEI Helios Nanolab 600 Dual-Beam using an automated routine designed and was executed with the Nano-Builder software.

The geometry of the DCB was optimized using an elastic finite element analysis in Ansys in order to magnify the stress at the bottom of the notch in comparison to the load that would cause fracture toward the top of each of the bending beams (see Supplementary Fig. 4).

FIB milling consisted of several stages all executed at 30 kV and with the sample placed normal to the FIB column. The current was reduced as the fabrication proceeded, roughing with currents of 21, 10, and 7 nA, to create a sufficiently large trench around the DCB and to reduce the cross-sectional area of the test DCB. Subsequently, milling was performed with a probe current of 1 nA to obtain the final cross-section dimensions and clean all the sidewalls. Using the same current, the sample was tilted by −1° and +1° to mill the bottom and top sidewalls, respectively, in order to reduce the taper. Once the rectangular sample was prepared, a trough was cut into the top at normal incidence at 1 nA. Finally, a notch was cut using a line scan at 10 pA to create a stress concentrator and initiate the crack. Between each step an image correlation alignment procedure was used to correct the beam shift due to the change of current or stage movement.

The final DCB geometry (Fig. 1d) had nominal dimension of between 10 and 15 μm height (*l*), 2 μm width (*2d*), 5 μm thickness (*t*), and other dimensions as outlined in Fig. 1d. Images were captured to record actual dimensions of each DCB for subsequent analysis. DCBs were fabricated on one of the single crystal portion

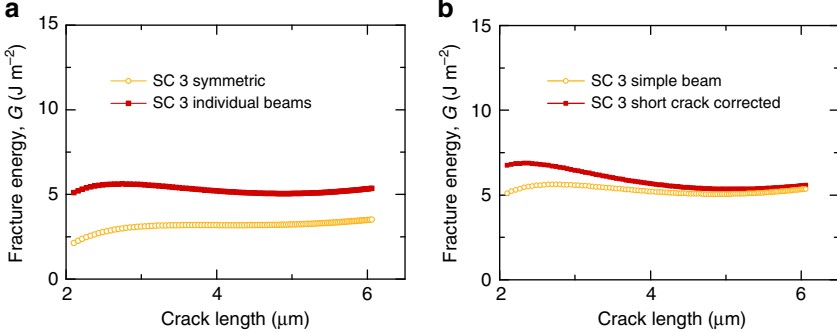

**Fig. 4** Effect of different assumptions on DCB geometry. **a** *Plot* showing how the fracture energy would be erroneously measured should the DCB tested be treated as perfectly symmetrical. For "SC 3 symmetric" the energy elastically stored in the system was measured by simply doubling the energy stored in one of the two beams. It is worth noting that the fracture energy could be underestimated or overestimated depending on whether the beam considered is moving more or less than the other beam, respectively. "SC 3 individual beams" were measured by measuring the energy stored in each beam. **b** "SC 3 simple beam" fracture energy is measured applying the simple beam theory assumptions, whereas the values in *red* are corrected to take into account the effect of shear when the crack is short

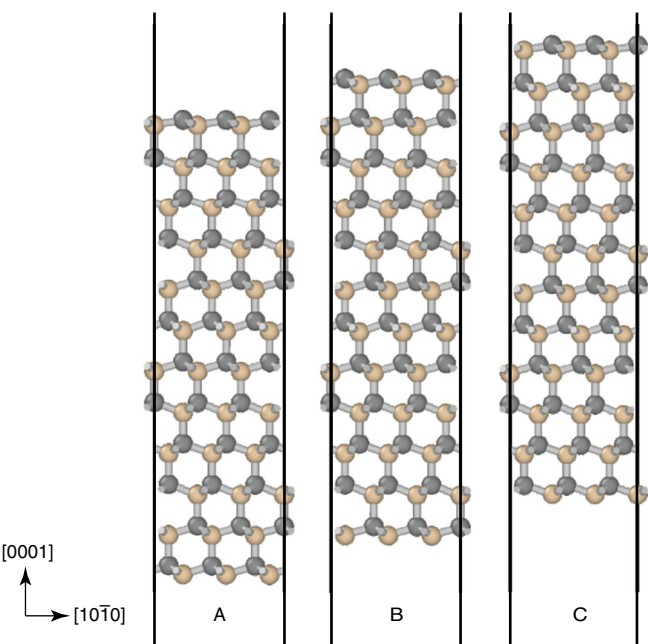

**Fig. 5** Relaxed structure of the 6H-SiC (0001) surface. There are three inequivalent basal cleavage planes for 6H SiC, which we find to be very close to degenerate in surface energy: $2\gamma = 8.58 \pm 0.04$ J m$^{-2}$ with LDA and $7.71 \pm 0.04$ J m$^{-2}$ with GGA. The upper surfaces are C-terminated and the lower Si-terminated

of the specimen with the notch aligned to have fracture on the basal plane with the crack propagating along the <a> direction.

For the bi-crystal, initial imaging of the sample surface was performed with a low current ion beam to locate the interface layer (see Supplementary Fig. 5) and the DCBs were fabricated using the same procedure as outlined for the single crystal sample. The interface was vertically running parallel to the <a> direction of the 6H-SiC crystal.

**Mechanical testing**. Mechanical testing was performed in situ in a SEM, which provided high spatial and temporal resolution imaging of the loading and fracture processes. This proved beneficial for the alignment of the sample and loading geometry, as well as direct observation of the fracture surface. Images were obtained using 5 kV with an InLens detector in an Auriga Zeiss SEM at low working distance (~5 mm). Video was recorded during test execution at a frame scan time of ~500 ms (using a reduced scan raster to balance dwell time, image quality, and fame rate). Frames so obtained had a pixel size of ~15 nm.

DCBs were tested with an Alemnis Nanoindenter. This was actuated by a piezoelectric transducer and, therefore, operates in displacement control. Testing

was performed using a 60° diamond wedge indenter of nominal tip length of 10 μm (Synton).

The nanoindenter is equipped with three-stage motors to move the sample with respect to its surface plane directions and the tip toward and away from the sample surface. We equipped the system with an additional rotational substage, over which the sample stub was mounted, in order to control with high precision the alignment between the notch and the wedge.

Displacement rates of between 1 and 2 nm s$^{-1}$ (Fig. 3a) were used to achieve a low crack propagation speed and to collect a high number of video frames for analysis. For the majority of tests, the displacement ramp was kept linear until the crack reached the bottom of the visible portion of the DCB and then the tip was retracted before causing complete failure of the DCB, for a test length of ~6 min. In three tests, after the crack had propagated a few microns into the DCB, the indenter was held in position with the DCBs still loaded for 2–5 min and until crack stability was observed.

**6H-SiC DFT calculations**. DFT calculations of the surface energy for the 6H polymorph of SiC were performed with the CASTEP code[28]. The crystal structure was obtained from experimental results by Capitani et al.[29] via the Inorganic Crystal Structure Database. The structure was oriented to open a (0001) surface, as in the mechanical test. Cleavage of this surface yields one Si-terminated and one C-terminated surface, which were both initially left unsaturated (passivation effects are discussed in more detail below).

Convergence tests were performed, leading to the selection of a $4 \times 4 \times 4$ k-point grid to sample the Brillouin zone of the bulk unit and a plane wave basis set truncated at a cutoff energy of 600 eV, together with the standard set of ultrasoft pseudopotentials distributed with CASTEP[28]. A self-consistent energy tolerance of $10^{-6}$ eV was used with finite basis set correction to account for changes in the cell size during geometry optimization.

The bulk unit cell was relaxed to within a force tolerance of 0.05 eV Å$^{-1}$ and a stress tolerance of 0.1 GPa, leading to the lattice parameters shown in Table 1a. To simulate the surfaces, vacuum was introduced in the (0001) direction. Geometry optimization of the surface cell was performed with a fixed lattice (Fig. 5). We verified that 10 Å is sufficient to decouple periodic images of the slab from one another. Tests with up to four layers along (0001) showed that two layers are sufficient. Calculations were performed with the local density approximation (LDA) and the PBE[30] parameterization of the generalized gradient approximation (GGA) to the exchange correlation functional.

Following relaxation, unreconstructed surface energies were computed using $\gamma = (E_{surface} - E_{bulk})/2\,A$, where $A$ is the cell area perpendicular to (0001), effectively averaging the Si-terminated and C-terminated surfaces. This is similar to the approach used by Leung et al. for 3C-SiC[22]. We confirmed that an alternative approach of isolating the Si-rich and C-rich surfaces in turn by passivating with hydrogen[31] leads to very similar results to the bare surface approach. We were not able to reproduce the numerical findings of ref. [31], where a significantly lower surface energy of $2\gamma = 4.5$ J m$^{-2}$ was reported. We attribute this to differences in the DFT code and pseudopotentials used; however, the consistency between our 3C surface energy calculations and ref. [22] and ref. [23] provides reassurance of the correctness of our approach.

We next investigated possible effects of surface reconstructions: while our study is the first to consider the 6H (0001) surface at the DFT level, this surface is structurally related to the more widely studied 3C (111) surface, which does have a number of known surface reconstructions. However, these reconstructions require either silicon adatoms[32] or the presence of additional environmental molecules such as $H_2$ or $O_2$[33]. Moreover, as brittle cleavage proceeds through thermodynamic energy balance, it is usually argued that the relevant surface energy is the as-cleaved

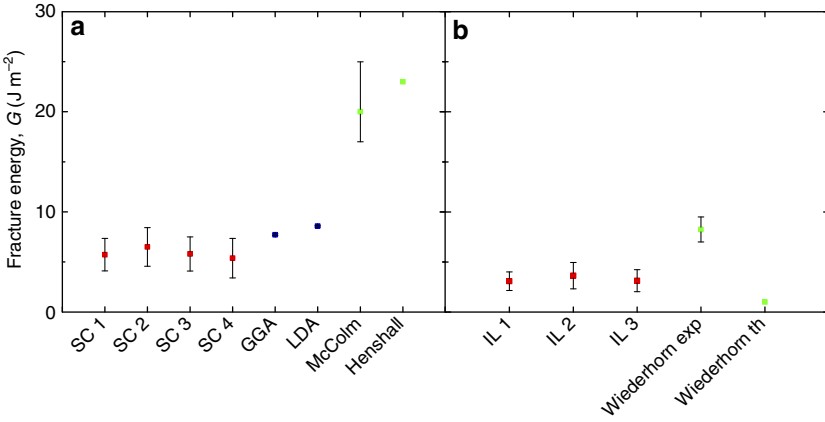

**Fig. 6** Comparison of data obtained in current work from experiments and DFT with existing literature on macroscopic tests. Data from current experimental work are measured as the average, while the error bar represents the standard error of mean, both as obtained from the Monte Carlo-based error propagation analysis (see Methods). Data from literature (McColm[37], Henshall[38], and Wiederhorn[26]) are represented as mean value, while the *error bar* represents the scatter. **a** Comparison of fracture/surface energy data from microscopic and macroscopic experimental testings and DFT calculations of 6H-SiC (0001) plane; **b** comparison of fracture energies measured on glassy interlayer of SiC bi-crystals with values obtained on glass from macroscopic tests and theoretical estimate

surface energy[17]. For a surface reconstruction to be relevant to determining the energy balance for fracture, it must form instantaneously from the propagating crack tip[34]. The geometry relaxations we performed allow us to exclude barrierless surface reconstructions.

A further complication is that the ABCACB layer stacking in 6H-SiC leads to three inequivalent basal cleavage planes (Fig. 5). However, we find that the different surface terminations are very close to degenerate in surface energy with both exchange correlation functionals, yielding $2\gamma = 8.58 \pm 0.04$ J m$^{-2}$ with LDA and $7.71 \pm 0.04$ J m$^{-2}$ with GGA.

From experimental considerations, we do not expect the fracture surface to be hydrogen-terminated prior to cleavage; moreover, there is unlikely to be sufficient time during fast fracture for stress-corrosion cracking to play a significant role. To be sure we computed DFT surface energies for a number of hydrogen-terminated SiC surfaces. We find surface energies in the range 0.1–0.5 J m$^{-2}$ depending on the chemical potential for hydrogen, significantly lower than the experimentally measured fracture energy, confirming that hydrogen termination is not relevant.

To assess the importance of entropic effects, suggested by Leung et al.[22] to lead to up to a 15% decrease in 3C–SiC surface energies at 300 K, we computed the surface energy using the quasi-harmonic approximation for two empirical force fields known to provide an accurate description of SiC[23], by subtracting the temperature-dependent free energy of the bulk configuration from that of the surface one. The frozen phonon supercell method was used to compute the force constant matrix at a number of volumes, together with Parlinksi-Li Fourier interpolation for the dispersion relations[35], as implemented in the phonopy code. Convergence of the surface free energy with respect to the number of supercells and the mesh used for integrating the entropic contribution over the Brillouin zone was obtained with a $2 \times 2 \times 2$ supercell and a $20 \times 20 \times 20$ $q$-point mesh. The results, illustrated in Supplementary Fig. 6, show a very small change in the surface free energy, corresponding to <4% decrease by 1000 K, and thus entropy can be expected to have only a very marginal effect on surface energies at room temperature.

Combining our results yields our best DFT estimate of the surface energy of base 6H SiC to be $2\gamma \sim 8.15 \pm 0.44$ J m$^{-2}$. Our error estimate includes contributions from the model error through the deviation between the LDA and GGA values ($\pm 0.43$ J m$^{-2}$) and the structural uncertainty arising from the inequivalent cleavage planes ($\pm 0.04$ J m$^{-2}$) as well as other sources of numerical error such as finite basis set error and numerical convergence ($<\pm 0.1$ J m$^{-2}$).

Finally, we calculated the relative fracture toughness values to facilitate comparison with literature presenting $K_{Ic}$ values. In order to do so for the DFT fracture energy data, we used the DFT elastic constants computed in a manner that is completely consistent with the DFT surface energies. The full $6 \times 6$ elastic constant matrix for 6H-SiC was computed with both LDA and GGA. Each independent elastic constant was determined from a linear fit to stress/strain data for five strains in the range −1.5 to 1.5%. Computing the compliance matrix $S = C^{-1}$ and propagating uncertainties from the fits lead to estimates of the elastic modulus perpendicular to the (0001) plane of $553 \pm 7.21$ GPa for LDA and $524 \pm 7.02$ GPa for PBE. Combining these elastic moduli with our surface energy calculations through $K_c = \sqrt{G_c E}$ produces the fracture toughness estimates shown in Table 1. Our combined DFT estimate $K_c = 2.10 \pm 0.08$ MPa m$^{0.5}$ compares well with a recent study that used DFT to model crack tip bond-breaking processes on the related (111) cleavage plane in 3C-SiC, where a fracture toughness of around 2.0 MPa m$^{0.5}$ was reported[36].

**Data availability**. Data for this paper are available in a Zenodo repository (doi: 10.5281/zenodo.398831).

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

## Acknowledgements

G.S., F.G., T.B.B., and D.B. acknowledge financial support from Element 6. T.B.B. has financial support for his fellowship from the Royal Academy of Engineering. We acknowledge financial support from EPSRC (EP/F033605/1, EP/L027682/1, EP/K028707/1, and EP/P002188/1). We thank Dr Rui Hao for fabrication of the the SiC bicrystals. We would like to thank Professor Eduardo Saiz, Drs Luc Vandeperre and Daniele Dini for helpful discussions on this work. An award of computer time was provided by the Innovative and Novel Computational Impact on Theory and Experiment (INCITE) program. This research used resources of the Argonne Leadership Computing Facility, which is a DOE Office of Science User Facility supported under Contract DE-AC02-06CH11357. Additional computing facilities were provided by the Centre for Scientific Computing of the University of Warwick with support from the Science Research Investment Fund.

## Author contributions

G.S., T.B.B., and F.G. planned the experiments and designed the data analysis approach; G.S. performed FIB sample fabrication and experiments; T.G. optimized the DCB sample geometry through FEA; D.S.B. derived the equation for fracture energy measurement; G.S., T.G., and T.B.B. developed the MATLAB script to analyze experimental data; P.P. and J.R.K. performed DFT calculations. All authors contributed to the final manuscript.

## Additional information

**Competing interests:** The authors declare no competing financial interests.

