## [Peer Review File · Nature Communications]

Reviewers' comments:

Reviewer #1 (Remarks to the Author):

The authors present thoroughly conducted new experiments at the micron scale which allow the in situ observation of a stably growing cleavage crack in a so called double cantilever beam geometry. For the first time SiC as a single crystal to validate the test setup and a diffusion bonded bi-crystal with a glassy interface were investigated. In the first case, the fracture energy of a specific cleavage plane was measured, whereas for the second case the fracture energy of the interface was tested and found to be significantly lower. For both materials, reproducible results were achieved which were then discussed in terms of the applied specimen geometry and furthermore compared to DFT calculations.

In my opinion the manuscript is well written but the scientific challenge claimed in the abstract was not sufficiently well addressed. The motivation for this work was to be able to test single grain boundaries at the micron scale. A reasonable test setup and methodology is therefore required. This aspect was elucidated nicely in detail in the manuscript. However, a single crystal or an artificial interface, as studied here, are nice test materials to validate the experimental approach but they could as well be tested at the macro-scale. Regarding the presented results, the question arises why a small scale mechanical test is needed at all. A grain boundary in a polycrystalline material was not studied. In our opinion this challenge needs to be elaborated more carefully to justify the high impact of this study. Furthermore, the DFT calculations accompanying the experiments cannot accurately reproduce the experimentally found values. It is unclear whether entropic contributions, as stated by the authors, are really responsible for the overestimation of the calculated DFT values. Further evidence would be needed here.

For the glassy interface the indicated range in which the experimental fracture energy falls, is also too large. Other brittle materials are situated in this range between 1...9.5 J/m². A macroscopic reference test could provide evidence whether the microscopically determined value is correct.

In the present form I recommend therefore to reject the paper, but I would encourage the authors to resubmit the results if they succeed in assessing grain boundary fracture toughness of a polycrystal using the same research methodology.

Here are a couple of questions/remarks concerning the article. The authors might find that helpful for further processing of their work:

1) Concerning p.4, line 66:

The authors claim that load control does not lead to stable fracture. Yet, this is strongly depending on the material and its crack resistance behavior.

2) Concerning p.4, line 73:

The fracture toughness should not be determined once a crack is growing. It is rather the initiation point of crack growth or the point of critical or unstable crack growth depending on

the sample/notch geometry.

3) Concerning p.5, lines 95-97:

Please add that the investigations were done for purely brittle materials.

4) Concerning p.6, line 110:

Explain/Indicate clearer: Where does equation (1) come from. The reference to three fracture mechanics books is too general.

5) Concerning p.7, line 124:

Explain/Indicate clearer: Where does the shear contribution in equation (2) come from. No reference is given.

6) Concerning p.9, line 175:

The fracture energy G was determined at which crack length? Could you also convert the critical energy release rate into a fracture toughness? If yes, which stress state would you expect in the samples?

7) Concerning p.9, line 177:

Why was the crack initiation site always so close to the machined notch? What is the influence of a crack which is created at the corner/bottom of one of the cantilevers or next to the interface. Can it be neglected?

8) Concerning p.11, line 205:

Would longer notches be of an advantage? What is the influence of the notch length?

9) Concerning p.11, line 215:

Can you explain what kind of effect longer beams, i.e. an increased dimension f , would have? Would the alignment be more difficult? Wouldn't longer beams be theoretically of an advantage because of the possible negligence of the shear component?

10) Concerning p.13, line 268:

The method was proposed for a straight cleavage plane and again a straight "artificial" interface. Can brittle grain boundaries which are in most cases not perfectly straight still be tested as the dimension l seems to have at least 10-15 μm ?

11) Concerning p.14, line 273:

Can you supply also an image of the bi-crystal sample – if possible with indication of the interface?

Reviewer #2 (Remarks to the Author):

This is unquestionably an interesting and well-written paper which presents an experimental method designed to generate stable crack growth data and fracture toughness at the micro-structural level of brittle materials like ceramics (i.e., at the micron scale). Certainly this will be of interest to material scientists specializing in ceramics, but the content and readability of the paper will attract a broader audience. My expertise is broadly in fracture and mechanics, but not specifically with in situ micron scale testing. The authors have cleverly taken over the wedge testing method long used at the macroscopic scale for their purposes at the scale of the micro-structure, using modern nano machining techniques to create the

test specimens and a nano-indenter to apply the loads (or displacement) to the specimen. I am not sufficiently familiar with in situ experimental techniques at this scale to know whether the authors are the first to this method of generating stable crack growth, but I'll take their word for it. The method clearly is successful.

The paper is also reasonably comprehensive in that it generates results for single crystal SiC and for a glass bonded interface between SiC crystals. Moreover, the team of authors carry out first principles calculation of the surface energy of the relevant SiC plane and provide a sound discussion of the theoretical predictions and theirs and others experimental measurements.

I think this paper meets the expectations for Nature Communications.

Reviewer #3 (Remarks to the Author):

Review of "In situ stable crack growth at the micron scale" by Sernicola et al.

I think this paper presents some interesting experiments concerning the propagation of cracks within the 6H-SiC. The experiments are then analyzed and compared with a continuum theory that is used to extract the surface energy relevant to cracking. These surface energies are then compared with DFT predictions. I was asked to comment on the DFT studies, so I will focus my attention on these calculations.

The DFT studies are carried out using CASTEP, a commercially available code, and employing two types of functionals. The authors have failed to indicate which type of pseudopotential (or PAW) they employ - this needs to be fixed. Otherwise, the convergence tests etc. seem reasonable.

I do, however, have issues with the analysis. First and foremost, the authors argue that based on reference 23, the surface free energies can vary with temperature by up to 40%. Reference 23 does, indeed, report this variation. However, the 40% reduction is observed at 1000 K, not 300 K. At 300 K, the reduction in energy is much closer to 15% resulting in much poorer agreement between experimental and measured surface energies.

Beyond this, I note that computing the surface energy of the 6H-SiC relevant to the fracture process is much less straightforward than this paper would have one believe. Because of the long period structure of the crystal normal to the basal plane, there are actually three different surfaces, depending on the termination plane. Why have the authors chosen the surface they have to study? Also, depending upon the atmosphere, the surfaces may or may not reconstruct (see for example Starke et al., Surface Review and Letters 6, 1129-1141, 1999). The vacuum within an SEM is not exceptional, and one expects some atmospheric effects. Do the surfaces reconstruct? Is the surface effectively hydrogen terminated? If so, then the surface calculations should reflect this. Also, the dynamics of opening a crack at the atomic scale are such that the surfaces left behind are far from perfect (see for example reference 23). What role do these imperfections play in the crack propagation? The paper, at a minimum, needs to address and dismiss these concerns before simply claiming good agreement between DFT and experiment.

As a result, I don't think that the DFT results here add much to the paper. The authors need to conduct a more thorough study including referencing the extensive literature on the subject of these surfaces to discern the materials science that is controlling the fracture process.

Reviewer #1 (Remarks to the Author):

The authors present thoroughly conducted new experiments at the micron scale which allow the in situ observation of a stably growing cleavage crack in a so called double cantilever beam geometry. For the first time SiC as a single crystal to validate the test setup and a diffusion bonded bi-crystal with a glassy interface were investigated. In the first case, the fracture energy of a specific cleavage plane was measured, whereas for the second case the fracture energy of the interface was tested and found to be significantly lower. For both materials, reproducible results were achieved which were then discussed in terms of the applied specimen geometry and furthermore compared to DFT calculations.

In my opinion the manuscript is well written but the scientific challenge claimed in the abstract was not sufficiently well addressed. The motivation for this work was to be able to test single grain boundaries at the micron scale. A reasonable test setup and methodology is therefore required. This aspect was elucidated nicely in detail in the manuscript.

The test has been designed to test at the micrometer lengthscale to enable testing of polycrystalline grain boundaries.

The impact of this study is that we have, for the first time, tested fracture in a stable crack geometry at the micrometer lengthscale through two cases: (1) the fracture energy of the basal plane in a SiC single crystal, and compared this to DFT simulations; (2) demonstrated potential in the technique to test specific interfaces, such as grain boundaries, through the glassy interface introduced in our bicrystal experiment.

To make this clearer we have amended the abstract.

However, a single crystal or an artificial interface, as studied here, are nice test materials to validate the experimental approach but they could as well be tested at the macro-scale. Regarding the presented results, the question arises why a small scale mechanical test is needed at all. A grain boundary in a polycrystalline material was not studied. In our opinion this challenge needs to be elaborated more carefully to justify the high impact of this study.

Testing of SiC at the macro-scale is limited in the literature. We found there is only one source using macroscopic testing techniques [1] and two using indentation fracture [2,3]. Values obtained in this prior work are markedly higher than both those stemming from theoretical calculations and found in our study. For instance, Henshall et al. [1] measured a fracture energy of 23 J m^{-2} from large scale tests. We agree with these authors who state that this is significantly higher than the surface energy particularly because they noticed a significant amount of roughness on the fracture surface. When using indentation [3,4] to measure the fracture toughness McColm [3] reports values between 17 and 25 J m^{-2} at different indentation loads for single crystal SiC. Again these are significantly higher than either the measured or calculated values reported here. Also it is worth noting the large scatter of the values obtained using these tests. Furthermore, similar problems have been encountered in the macroscopic testing of other single crystals of brittle materials [5]. These issues and

discrepancies indicate that there is value, novelty and importance in testing directly at the small scale.

Case 2, on the bicrystal:

Producing a bicrystal with a homogeneous interface of 10 nm along a sample of several tens of millimetres is very difficult. Should it be possible to make the sample, then the same challenges of clean fracture as the single crystal arise.

In conclusion, we believe that in our manuscript we clearly show that the values obtained with our approach provide complimentary information not available through macroscopic testing even on these simple geometries.

To make the distinction clearer we have included the earlier data on plots of the results from both the single crystals and interface samples in the manuscript fig.6.

We agree with the reviewer that testing grain boundaries in polycrystalline materials is the next important challenge to tackle and that is the focus of our next work in this area. This is now referred to in the discussion section of the manuscript

Furthermore, the DFT calculations accompanying the experiments cannot accurately reproduce the experimentally found values. It is unclear whether entropic contributions, as stated by the authors, are really responsible for the overestimation of the calculated DFT values. Further evidence would be needed here.

Regarding the DFT calculations, we refer the referee to our detailed response to Referee 3 below, which includes new calculations. We agree with reviewer 1 that entropy is not a major contributor.

Following the comments from the reviewer we have rechecked our calculations and found an error relating to how the elastic modulus was calculated for the correct orientation. We need two stiffness values for our analysis: (1) $E // (0001)$ to calculate the fracture energy for beam bending; (2) $E \perp (0001)$ for conversion of the fracture energy to toughness (and note that SiC is transversely isotropic). We have used the values of elastic constants from Landolt-Börnstein [6], and find a value of 480 GPa for the direction parallel to the (0001) plane.

Furthermore we have now extracted the elastic moduli and Poisson's ratio values directly from the DFT- GGA and LDA calculations.

As mentioned above we have now converted the fracture energy values to fracture toughness, K_{Ic} values, using the respective elastic modulus perpendicular to the (0001) plane and find values as summarised in the table below.

Elastic modulus // (0001), E ; [GPa]	Poisson's ratio ν_{12}	Fracture energy*, G_c or surface energy**, 2γ ; [J m ⁻²]	Elastic modulus \perp (0001), E ; [GPa]	Fracture toughness, K_{Ic} ; [MPa m ^{1/2}]
480	0.180	measured from DCB 5.95 ± 1.79 (*)	554	1.80±0.26

		calculated by DFT-GGA		
475	0.179	7.71 ± 0.04 (**)	524±7.02	2.01±0.03
		calculated by DFT-LDA		
499	0.190	8.58 ± 0.04 (**)	553±7.21	2.18±0.03
		combined DFT - LDA & GGA		
487	0.184	8.15 ± 0.44	538±5.03	2.10±0.08

Our experiments show that the experiments and simulations are converging, as compared to previous work, but there are still differences of $\sim 2 \text{ Jm}^{-2}$ or $\sim 0.3 \text{ MPa m}^{0.5}$. We are unsure whether this difference is related to issues in the experiment or simulation (as we have now propagated obvious errors in both). We feel that it is appropriate for us to report our work, as a ‘best in show’ example, and frankly discuss these differences in the discussion.

These findings did prompt us to propagate errors in our analysis, and we used a Monte-Carlo based approach. Gaussian distributions of each critical measurement were introduced with reasonable standard deviations based upon experimental observations. The modified values of each variable were introduced, with independent additions of noise, into the analysis and the calculations were made to calculate the final values of G. The error bars for G reflect the standard deviation of 1000 of these MC tests.

This is introduced within the text and the distributions of error for each variable are reported in the supplementary data.

For the glassy interface the indicated range in which the experimental fracture energy falls, is also too large. Other brittle materials are situated in this range between 1...9.5 J/m². A macroscopic reference test could provide evidence whether the microscopically determined value is correct.

The large range referred to in the manuscript is as reported by Wiederhorn [7] in a review on the macroscopic fracture toughness of glass. In this work the author notes that the values obtained from large scale samples testing are too high when compared to theoretical calculations of surface energy, and justifies the findings with the difficulties in obtaining a straight crack in this material. The suggestion of the reviewer of repeating such a macroscopic test would result in a similar range of values, and fabricating such a sample of length of 10s of millimetres with an interface of $\sim 10\text{nm}$ is physically very difficult.

In the present form I recommend therefore to reject the paper, but I would encourage the authors to resubmit the results if they succeed in assessing grain boundary fracture toughness of a polycrystal using the same research methodology.

We share the feelings of Reviewer #2 that our contribution presents a “comprehensive” work on “an experimental method designed to generate stable crack growth data and fracture toughness at the micro-structural level of brittle materials” together with a “a sound discussion of the theoretical predictions and theirs and others experimental measurements”. This provides a significant leap forward in experimental design and promotes a new understanding of the fracture behaviour of individual interfaces, which are a necessary step in the quest to testing boundaries in polycrystalline materials.

This is now referred in the discussion section of the manuscript.

Here are a couple of questions/remarks concerning the article. The authors might find that helpful for further processing of their work:

1) Concerning p.4, line 66:

The authors claim that load control does not lead to stable fracture. Yet, this is strongly depending on the material and its crack resistance behavior.

We claim that: “load control inherently does not lend to stable fracture”. In cases where there is local plasticity, the crack may be stabilised by the plastic zone ahead of the crack tip, but in brittle materials this load control results in unstable crack growth. **We have updated the manuscript text to make this clearer:**

2) Concerning p.4, line 73:

The fracture toughness should not be determined once a crack is growing. It is rather the initiation point of crack growth or the point of critical or unstable crack growth depending on the sample/notch geometry.

This is only true for unstable crack growth (where the energy released is not in equilibrium with the fracture energy). We have stable crack growth and, as demonstrated, and can stop the crack as needed. This is the inherent novelty in our microscale test, where our progressive measurement is therefore a fracture toughness measurement. In the existing literature, this has been widely taken advantage of at the macroscale, e.g. as shown by Wegst et al. [8].

To clarify this point for other readers, we have changed the term fracture toughness to fracture energy to both avoid confusion and be consistent with what is measured in the paper.

3) Concerning p.5, lines 95-97:

Please add that the investigations were done for purely brittle materials.

We have changed the text according to the reviewer’s suggestion.

4) Concerning p.6, line 110:

Explain/Indicate clearer: Where does equation (1) come from. The reference to three fracture mechanics books is too general.

Following the suggestion we have changed the text to make it clearer that the equation is that for bending of an Euler-Bernoulli beam.

5) Concerning p.7, line 124:

Explain/Indicate clearer: Where does the shear contribution in equation (2) come from. No reference is given.

We derived Eq. (3) ourselves starting from the considerations of Timoshenko’s beam theory.

Derivation of this equation has been added as supplementary information.

6) Concerning p.9, line 175:

The fracture energy G was determined at which crack length? Could you also convert the critical energy release rate into a fracture toughness? If yes, which stress state would you expect in the samples?

The value presented in the table is the average of all the fracture energy measurement obtained with crack evolution for a specimen. Our samples fracture in mode I. The well-known relation between energy release rate and stress intensity factors $G_c = K_c^2/E$, valid in mode I fracture under plane stress conditions, allows the conversion required. This was used to calculate the values in the table above. Given the small value of Poisson's ratio in this case (of order 0.08, smaller than the values in the table above which are for loading parallel to the basal plane) the difference between plane stress and plane strain assumptions is negligible.

7) Concerning p.9, line 177:

Why was the crack initiation site always so close the machined notch? What is the influence of a crack which is created at the corner/bottom of one of the cantilevers or next to the interface. Can it be neglected?

We machined the notch to create a stress concentration (as mentioned at p.15, line 299) in order to have the crack initiation as close to the centre of the DCB as possible.

As we want to test the fracture energy of the interface, we have optimized the geometry to have the crack run along this. Therefore, if the crack was to grow elsewhere we would discard the test.

We have clarified this in the text:

8) Concerning p11, line 205:

Would longer notches be of an advantage? What is the influence of the notch length?

The initial notch selects where the crack generates and the stubbiness correction manages the shear contribution of the beam geometry. The stable measurement of G as a function of time indicates that we are relatively notch-length insensitive.

9) Concerning p.11, line 215:

Can you explain what kind of effect longer beams, i.e. an increased dimension f , would have? Would the alignment be more difficult? Wouldn't longer beams be theoretically of an advantage because of the possible negligence of the shear component?

We are designing a test to be suitable to test real grain boundaries, so we desire a short beam. We have introduced the Timoshenko analysis to include the effect of the shear in the beam.

We agree that a longer beam would be useful in our example for exemplar measurements but this is not that important in a practical sense.

10) Concerning p.13, line 268:

The method was proposed for a straight cleavage plane and again a straight "artificial" interface. Can brittle grain boundaries which are in most cases not perfectly straight still be tested as the dimension l seems to have at least 10-15 μm ?

The stubbiness correction, including shear, enables suitable measurement of G for less than 1 micrometer from the pre-notch (see Figure 4.(d) in the manuscript), as the crack does not have to

extend along the entire length of the DCB in this geometry. This is promising as we want to extend this test to cracks in less ideal bi-crystals (as found in polycrystalline materials).

11) Concerning p.14, line 273:

Can you supply also an image of the bi-crystal sample – if possible with indication of the interface?

It is difficult to image the interface because its thickness is comparable to the resolution of the SEM. In addition, since the atomic masses between SiC and SiO₂ are similar they do not show a good contrast in backscatter mode. However after the sample has been FIB'd due to preferential etching the interface can be seen running through the centre of the sample as in the image below (pointed by the arrow). **This image has been added to the supplementary information.**

Reviewer #2 (Remarks to the Author):

This is unquestionably an interesting and well-written paper which presents an experimental method designed to generate stable crack growth data and fracture toughness at the micro-structural level of brittle materials like ceramics (i.e., at the micron scale). Certainly this will be of interest to material scientists specializing in ceramics, but the content and readability of the paper will attract a broader audience. My expertise is broadly in fracture and mechanics, but not specifically with in situ micron scale testing. The authors have cleverly taken over the wedge testing method long used at the macroscopic scale for their purposes at the scale of the micro-structure, using modern nano machining techniques to create the test specimens and a nano-indenter to apply the loads (or displacement) to the specimen. I am not sufficiently familiar with in situ experimental techniques at this scale to know whether the authors are the first to this method of generating stable crack growth, but I'll take their word for it. The method clearly is successful.

The paper is also reasonably comprehensive in that it generates results for single crystal SiC and for a glass bonded interface between SiC crystals. Moreover, the team of authors carry out first principles calculation of the surface energy of the relevant SiC plane and provide a sound discussion of the theoretical predictions and theirs and others experimental measurements.

I think this paper meets the expectations for Nature Communications.

We thank this Referee for their positive and encouraging remarks on our work.

Reviewer #3 (Remarks to the Author):

Review of "In situ stable crack growth at the micron scale" by Sernicola et al.

I think this paper presents some interesting experiments concerning the propagation of cracks within the 6H-SiC. The experiments are then analyzed and compared with a continuum theory that is used to extract the surface energy relevant to cracking. These surface energies are then compared with DFT predictions. I was asked to comment on the DFT studies, so I will focus my attention on these calculations.

The DFT studies are carried out using CASTEP, a commercially available code, and employing two types of functionals. The authors have failed to indicate which type of pseudopotential (or PAW) they employ - this needs to be fixed. Otherwise, the convergence tests etc. seem reasonable.

We used the standard set of ultrasoft pseudopotentials distributed with CASTEP for all calculations reported in this work. We thank the Referee for spotting this omission, which is corrected in the revised manuscript.

I do, however, have issues with the analysis. First and foremost, the authors argue that based on reference 23, the surface free energies can vary with temperature by up to 40%. Reference 23 does, indeed, report this variation. However, the 40% reduction is observed at 1000 K, not 300 K. At 300 K, the reduction in energy is much closer to 15% resulting in much poorer agreement between experimental and measured surface energies.

We agree with the Referee that this is a critical point and we apologise for misreporting the results of Ref. 23 in our original submission (listed as [10] here). Motivated by this, and with the aim of obtaining a detailed understanding of what is going on in our fracture experiments, we have carried out free energy calculations for the SiC 6H surface and bulk unit cells using two empirical force fields [9]. Results using both harmonic and quasi-harmonic approximations are illustrated below. Full details have been added to the methods section in the revised manuscript, along with a new

Supplementary Figure 6.) We find a smaller change in the surface free energy than the authors of [10] for 3C SiC, corresponding to <4% decrease by 1000 K and thus only a very marginal effect at 300 K.

While carrying out free energy simulations of SiC 6H supercells at the DFT level would be possible, the empirical calculations presented here agree with the Referee's hypothesis that a significant decrease in surface energy at 300 K is unlikely. We believe that a free energy DFT study would provide only marginal additional insight.

Fig 1. Surface energies 2γ of the SiC 6H (0001) surface as a function of temperature using the harmonic and quasi-harmonic approaches. Screened versions of the Tersoff and Erhart & Albe interatomic potentials have been used [9]. For both models the reduction in surface energy at 1000 K is <4%.

The **Methods section in the revised manuscript includes a more detailed review of the existing literature** regarding surface energy calculations for SiC. In conducting this updated review, we encountered an apparent contradiction: Ref. 11 reports SiC 3C surface energies of $2\gamma \sim 4.5 \text{ J/m}^2$, which is much lower than our results or those of Refs. 9 and 10 and our own DFT calculations (for either the 6H or 3C surfaces). Abavare et al. [11] note that for polar surfaces, long range interactions can make surface energy convergence with system size very slow, and propose instead H-terminating one of the two surfaces and using reference chemical potentials from silane and methane molecules to compute the surface energy of the other (unterminated) surface.

We followed the same procedure for both 3C and 6H, but were unable to replicate their results. For the 6H surface we find very similar surface energies to those we had previously obtained using two bare surfaces. We also computed the 3C surface energy with the same GGA-PBE functional as Ref. [11], obtaining $2\gamma = 8.42 \text{ J/m}^2$, which is also much larger than the value obtained by Abavare et al. Possible explanations are differences in pseudopotentials and DFT code used. However, the

consistency with our own previous results and with those of Refs. [9] and [10] provides reassurance.

SiC	Surface Passivation	DFT (GGA-PBE) Surface Energy 2γ / J/m ²
6H	None	7.71
	One H-passivated	7.26
3C	One H-passivated	8.42

Beyond this, I note that computing the surface energy of the 6H-SiC relevant to the fracture process is much less straightforward than this paper would have one believe. Because of the long period structure of the crystal normal to the basal plane, there are actually three different surfaces, depending on the termination plane. Why have the authors chosen the surface they have to study?

We agree with the Referee that there are many further details that may be important when computing surface energies. We looked at some of these in preparatory calculations not reported in our initial submission, but accept that the way in which the calculations were presented in the initial submission made the process appear overly simplistic; this was in part a deliberate choice to avoid distracting from the novel experimental work. To address this comment, we have significantly extended the Discussion and Methods sections in the revised manuscript, and included results from a number of new calculations, as outlined below.

Regarding the choice of termination plane, for 6H SiC the ABCACB layer stacking leads to three inequivalent basal cleavage planes as the Reviewer points out. The table below shows results for 2γ in the static limit for each of these three terminations, in J/m².

Termination	Exchange Correlation Functional (J/m ²)	
	LDA	GGA (PBE)
A	8.51	7.65
B	8.68	7.81
C	8.55	7.68

Mean	8.58	7.71
Standard error	0.04	0.04

These three choices are very close to degenerate in surface energy with both XC functionals, with a standard error in the mean of less than 0.05 J/m^2 . We have added a sentence to the Methods section in the revised manuscript, and amended Figure 2 to illustrate the three cleavage planes.

Fig 2. The three inequivalent basal cleavage planes for 6H SiC, which are very close to degenerate in surface energy: $2\gamma = 8.58 \pm 0.04 \text{ J/m}^2$ with LDA and $7.71 \pm 0.04 \text{ J/m}^2$ with GGA.

Also, depending upon the atmosphere, the surfaces may or may not reconstruct (see for example Starke et al., Surface Review and Letters 6, 1129-1141, 1999). The vacuum within an SEM is not exceptional, and one expects some atmospheric effects. Do the surfaces reconstruct?

Whether or not surface reconstruction is relevant to determining the fracture energy is a matter of some contention in the literature on atomistic modelling of fracture. Since brittle cleavage proceeds through thermodynamic energy balance, it is usually argued that the relevant surface energy is the as-cleaved surface energy. For a surface reconstruction to be relevant to determining the energy balance for fracture, it must form instantaneously from the propagating crack tip [12]. In practice this means that as well as being energetically favourable compared to the cleaved surface, the barrier to form the reconstruction must be significantly smaller than $k_B T$, so that the surface forms immediately. The geometry relaxations we performed allow us to exclude simple reconstructions which form spontaneously without a barrier.

We are not aware of any extensive studies of surface reconstructions of 6H SiC (0001) at the DFT level. The 6H (0001) is structurally closely related to the more widely studied 3C (111) surface, which does have a number of known reconstructions. However, these reconstructions require either silicon adatoms [13] or the presence of additional environmental molecules such as H₂ or O₂ (Ref. [14], see also discussion below), indicating they are unlikely to be relevant here. **We have extended the discussion in the manuscript to reflect this.**

In summary, after a number of careful checks we are confident that the best DFT estimate of the surface energy of 6H SiC is of the order $2\gamma \sim 8.15 \pm 0.44 \text{ J/m}^2$. This error estimate includes contributions from the model error through the deviation between the LDA and GGA values ($\pm 0.43 \text{ J/m}^2$) and the structural uncertainty arising from the inequivalent cleavage planes (± 0.04) as well as other sources of numerical error such as finite basis set error and numerical convergence ($<\pm 0.1 \text{ J/m}^2$).

Is the surface effectively hydrogen terminated? If so, then the surface calculations should reflect this.

From experimental considerations we do not expect a significant concentration of hydrogen in the chamber; however to be sure we extended the passivation approach described above to compute DFT surface energies for a number of hydrogen terminated surfaces. We find surface energies in the range 0.1-0.5 J/m² depending on the chemical potential for hydrogen, which is itself a function of the concentration of hydrogen and whether it is in atomic or molecular form. **We have added a brief description of these calculations to the Methods section in the revised manuscript.**

These values are significantly lower than the experimentally measured fracture energy of $\sim 6 \text{ J/m}^2$, confirming our hypothesis that hydrogen termination is not relevant here. This is consistent with our argument above regarding reconstructions that the surface energy that is relevant for the thermodynamic energy balance for cleavage is that of the bare surface, i.e. the crystal will first cleave before later potentially becoming H-saturated, as there is unlikely to be sufficient time during fast fracture for stress-corrosion cracking to play a significant role.

In addition, our experiments included a stage during which the wedge was held in position for up to ~ 5 minutes. Analysis of the frames during such period show no observable crack extension and, consequently, no significant change in the measured fracture energy. This is a strong indication that there are no measurable slow crack effects

Also, the dynamics of opening a crack at the atomic scale are such that the surfaces left behind are far from perfect (see for example reference 23). What role do these imperfections play in the crack propagation? The paper, at a minimum, needs to address and dismiss these concerns before simply claiming good agreement between DFT and experiment.

As a result, I don't think that the DFT results here add much to the paper. The authors need to conduct a more thorough study including referencing the extensive literature on the subject of these surfaces to discern the materials science that is controlling the fracture process.

To address concerns of this Reviewer, we have expanded and improved the DFT contribution to this paper. Key contributions of the DFT work:

- (1) Our calculations are the first DFT studies of the 6H surface.
- (2) They allow us to exclude effects of reconstruction and H-passivation on the fracture energy.
- (3) The DFT studies are complementary and provide a multidisciplinary approach to studying fracture, and the close (but not perfect) agreement of simulations and experiment is of value to the experimental and simulation communities.

We agree that the impact of surface imperfections is of significant interest. We could imagine a significant manuscript, motivated by the differences found in this paper, which extends our work. This would consider dynamic contributions, that require fully dynamic simulations with a combined quantum-mechanical description of bond-breaking, similar to prior work in silicon [15] and beyond the approach in [10]. We anticipate that such a study would improve our understanding of the detailed fracture morphology, and perhaps could refine predictions and explain the $0.3 \text{ MPa}^{0.5}$ difference between our calculations and experiment.

References

1. Henshall, J. L., Rowcliffe, D. J. & Edington, J. W. Fracture Toughness of Single-Crystal Silicon Carbide. *J. Am. Ceram. Soc.* 60, 373–375 (1977).
2. Adewoye, O. O. Surface and subsurface fracture in single crystal of alpha-SiC. in *Fracture Mechanics of Ceramics Vol. 5* (ed. R.C. Bradt, A.G. Evans, D.P.H. Hasselman, F. F. L.) 107–120 (Plenum Press, 1983).
3. McColm, I. J. *Ceramic Hardness*. (Springer US, 1990). doi:10.1007/978-1-4757-4732-4
4. Henshall, J. L. & Brookes, C. A. The measurement of K_{IC} in single crystal SiC using the indentation method. *J. Mater. Sci. Lett.* 4, 783–786 (1985).
5. Wiederhorn, S. M. Fracture of Sapphire. *J. Am. Ceram. Soc.* 52, 485–491 (1969).
6. Landolt, H. & Börnstein, R. *Elektrische, Piezoelektrische, Pyroelektrische, Piezooptische, Elektrooptische Konstanten und Nichtlineare Dielektrische Suszeptibilitäten*. Zahlenwerte und Funktionen aus Naturwissenschaft und Technik. Vol. Gruppe III, (Springer-Verlag, 1979). 6.6, 7.
7. Wiederhorn, S. M. Fracture Surface Energy of Glass. *J. Am. Ceram. Soc.* 52, 99–105 (1969).
8. U.G.K. Wegst, H. Bai, E. Saiz, A.P. Tomsia, R.O. Ritchie, Bioinspired structural materials, *Nat Mater.* 14 (2015) 23–36. <http://dx.doi.org/10.1038/nmat4089>.
9. L. Pastewka, A. Klemenz, P. Gumbsch, and M. Moseler, *Phys. Rev. B* **87**, 205410 (2013).
10. K. W. K. Leung, Z. L. Pan, and D. H. Warner, *Acta Mater.* **77**, 324 (2014).
11. E. K. K. Abavare, J.-I. Iwata, A. Yaya, and A. Oshiyama, *Phys. Status Solidi* **251**, 1408 (2014).
12. D. Fernandez-Torre, T. Albaret, and A. De Vita, *Phys. Rev. Lett.* **105**, 185502 (2010).
13. J. Pollmann and P. Krüger, *J. Phys. Condens. Matter* **16**, S1659 (2004).
14. U. Starke, J. Bernhardt, J. Schardt, and K. Heinz, *Surf. Rev. Lett.* **06**, 1129 (1999).
15. J. R. Kermode, L. Ben-Bashat, F. Atrash, J. J. Cilliers, D. Sherman, and A. De Vita, *Nat. Commun.* **4**, 2441 (2013).

Reviewers' comments:

Reviewer #1 (Remarks to the Author):

The manuscript was revised in detail by the authors and I suggest now that it can be published with minor revisions. Minor revisions are required due to the many corrections made, and a number of new open questions arise, which need to be addressed before publishing the work:

1) The DFT calculations agree now well with the experimental data. Based on the comments made by reviewer #3, it should be explained now what can be learnt from this agreement. What is DFT exactly useful for in the context of this study? Which factors lead to the change in fracture behavior compared to macroscopic tests? The authors claim that a new understanding of the fracture behavior of individual interfaces was promoted. Which new understanding is it exactly and can it be explained by DFT?

2) As explained on page 2 of the answers: There is an obvious difference in the fracture energy between macroscopic tests and small-scale tests in SiC. Macroscopic values are in the order of 17-25 J/m², whereas values of 6-8 J/m² are reported at the micro-scale. This is a factor of three. Do the authors now claim here that there is a size effect related to fracture toughness? The authors should comment on this.

3) Why do the authors extract the elastic constants from the DFT calculations and why are the values not consistent? Size effects of Young's moduli and Poisson's ratio should only be expected for sample sizes below 10 nm, when the number of the surface atoms becomes significant. This is not the case in the experiments the authors present. Therefore tabulated values in reference databases such as Landolt-Börnstein are for sure more accurate than any DFT calculation can be. The authors should comment on this.

Reviewer #3 (Remarks to the Author):

The authors have conducted an extensive study based on all the reviewer's comments. They are to be applauded for their thorough study. I think that the paper is much improved, and I recommend publication in Nature Communications.

We thank both Reviewers for their positive and encouraging remarks on our work, which we wish to resubmit for consideration for publication in Nature Communications. We include below a detailed point by point response to the questions raised by Reviewer #1, along with a summary of the changes we have made to the manuscript.

Reviewers' comments:

Reviewer #1 (Remarks to the Author):

The manuscript was revised in detail by the authors and I suggest now that it can be published with minor revisions. Minor revisions are required due to the many corrections made, and a number of new open questions arise, which need to be addressed before publishing the work:

1) The DFT calculations agree now well with the experimental data. Based on the comments made by reviewer #3, it should be explained now what can be learnt from this agreement. What is DFT exactly useful for in the context of this study?

We suggest that this is addressed within the manuscript in the following places, and in particular, note that unlike prior studies fracture tests which do not measure the surface energy, our study agrees with DFT calculations:

Quoting from the abstract (lines 19-21):

"These experiments correlate well with our density functional theory calculations of the surface energy of the same 6H-SiC basal cleavage plane"

And from discussion in our manuscript (lines 268-272):

"The method was validated by fracturing single crystal 6H-SiC along the  direction, finding a value of the fracture energy of 5.95 ± 1.79 J/m². DFT calculations were performed on the same crystallographic plane as experimentally tested for the first time. While the calculated values were slightly higher than the experimental values, both measurements agree well when compared to previous measurements (Fig 6)."

We have articulated this in our prior response to reviewers, in comments to reviewer #3 in the 1st round of comments:

"Key contributions of the DFT work:

- (1) Our calculations are the first DFT studies of the 6H surface.*
- (2) They allow us to exclude effects of reconstruction and H-passivation on the fracture energy.*
- (3) The DFT studies are complementary and provide a multidisciplinary approach to studying fracture, and the close (but not perfect) agreement of simulations and experiment is of value to the experimental and simulation communities."*

Moreover, the consistency between experiment and DFT fracture energies for the single crystal case gives us confidence in the approach when moving to study more complex interfaces. We have added a remark to the manuscript to make these points clear to all readers.

Which factors lead to the change in fracture behavior compared to macroscopic tests?

We observe sharp and stable crack growth with:

- no crack deflection (due to defects, voids, stress misalignments),
- a sharp crack (i.e. limited plastic dissipation),
- reduced environmental effects.

In prior tests either there was crack deflection, cracks were not sharp, or there were environmental effects.

We have addressed this in our response to reviewer #1 in the 1st round of comments :

"For instance, Henshall et al. [1] measured a fracture energy of 23 J m^{-2} from large scale tests. We agree with these authors who state that this is significantly higher than the surface energy particularly because they noticed a significant amount of roughness on the fracture surface."

This is highlighted in our manuscript (lines 242-247):

"From the high resolution SEM images the crack propagation in the single crystal SiC appears to create two new smooth edges with no evident deviation from a linear path. The crack tip is sharp and no plastic deformation is expected around it and likewise no toughening mechanisms are available in the material examined. Furthermore, it is worth noting that the energy values were measured far from the notch and so we do not expect to be affected by crack initiation processes or ion penetration damage, which is found to be $<100 \text{ nm}$ from the surface^{24,25}."

The authors claim that a new understanding of the fracture behavior of individual interfaces was promoted. Which new understanding is it exactly and can it be explained by DFT?

We draw the reviewer to consider lines 46-51 in our manuscript:

"Macroscopic tests are very successfully used to understand bounds on component performance and to compare and contrast ceramic processing strategies, but as they test polycrystalline aggregates it is very difficult to understand the role of specific microstructural features. Achieving a step change in the performance and utilization of advanced ceramics requires new insight at the microstructural lengthscale, as failure of ceramics is controlled often by the weakest microstructural link."

And lines 72-80 in our manuscript:

"In light of these previous geometries⁸, it would be ideal to have manufactured samples with geometrical features enabling:

- stable crack growth beyond any damaged region, in order to measure fracture toughness as the crack evolves and to overcome limitations imposed by FIB-induced damage;*
- relative freedom in the positioning of the notch;*
- minimization of the effect of frame compliance and friction between the indenter and the sample, making evaluation of the measured energy easier.*
- use of a relatively simple sample geometry thus facilitating sample fabrication and fracture or surface energy analysis;"*

ultimately leading to lines 96-98 in our manuscript:

"In summary, this work presents a novel method to measure the fracture energy evolution with crack length at the microscale in purely brittle materials, with the benefits of a high sampling rate, stable crack growth and a controlled environment"

This is then contextualised in our Discussion within our manuscript (lines 283-286):

"The method proposed introduces a new opportunity to study the fracture properties at the lengthscale of individual interfaces and microstructural units, thus opening up to the possibility of optimization of fracture properties of specific boundaries in ceramics with more complex microstructures."

The contribution made by our DFT studies is addressed in our response to the Reviewer's first point above.

2) As explained on page 2 of the answers: There is an obvious difference in the fracture energy between macroscopic tests and small-scale tests in SiC. Macroscopic values are in the order of 17-25 J/m², whereas values of 6-8 J/m² are reported at the micro-scale. This is a factor of three. Do the authors now claim here that there is a size effect related to fracture toughness? The authors should comment on this.

We do not claim or believe the discrepancy is due to a size effect related to fracture energy. We note that through testing at the small scale, we obtain sharp and straight cracks, which in absence of any evidence of dissipation due to bridging, deflection or other mechanisms, agrees well with values of the surface energy computed with DFT.

3) Why do the authors extract the elastic constants from the DFT calculations and why are the values not consistent? Size effects of Young's moduli and Poisson's ratio should only be expected for sample sizes below 10 nm, when the number of the surface atoms becomes significant. This is not the case in the experiments the authors present. Therefore tabulated values in reference databases such as Landolt-Börnstein are for sure more accurate than

any DFT calculation can be. The authors should comment on this.

We agree with reviewer #1 that Landolt-Börnstein's values are very reliable and for this reason we used them for our experimental energy measurements (see lines 180-186 from our manuscript).

In addition to the fracture energy values, we calculated the relative fracture toughness values to facilitate comparison with literature presenting K_{Ic} values. In order to do this for the DFT fracture energy data we felt it was important to use DFT elastic constants computed in a manner that is completely consistent with the DFT surface energies.

It should be noted that we believe the values are quite consistent and vary only by a few percent, as reported in the table below from the previous response to reviewers in the 1st round of comments. Furthermore, the effect on the K_{Ic} of such differences is negligible ($<0.05 \text{ MPa m}^{1/2}$).

Extracted from	Elastic modulus // (0001), E ; [GPa]	Elastic modulus \perp (0001), E ; [GPa]	Poisson's ratio ν_{12}
Landolt-Börnstein	480	554	0.180
DFT-GGA	475	524 ± 7.02	0.179
DFT-LDA	499	553 ± 7.21	0.190
combined DFT-LDA & GGA	487	538 ± 5.03	0.184

Difference for $E_{//}(0001)$ between:

- DFT-GGA and Landolt-Börnstein = -1%
- DFT-LDA and Landolt-Börnstein = +4%
- combined DFT-LDA & GGA and Landolt-Börnstein = +1.5%

Difference for $E_{\perp}(0001)$ between:

- DFT-GGA and Landolt-Börnstein = -5.4%
- DFT-LDA and Landolt-Börnstein = -0.2%
- combined DFT-LDA & GGA and Landolt-Börnstein = -2.9%

We have added a remark to the manuscript to make this clear.

Reviewer #3 (Remarks to the Author):

The authors have conducted an extensive study based on all the reviewer's comments. They are to be applauded for their thorough study. I think that the paper is much improved, and I recommend publication in Nature Communications.

We are grateful to reviewer #3 for the constructive remarks offered in the first round and for this encouraging comment.

REVIEWERS' COMMENTS:

Reviewer #1 (Remarks to the Author):

I'm happy now with the last version. "Fair enough" also the answers of the authors to my questions. I don't suggest further revisions. I recommend publishing the paper.

REVIEWERS' COMMENTS:

Reviewer #1 (Remarks to the Author):

I'm happy now with the last version. "Fair enough" also the answers of the authors to my questions. I don't suggest further revisions. I recommend publishing the paper.

We thank reviewer #1 for the positive feedback on our manuscript.